# Process Window and Repeatability of Thermomechanical Tangential Ring Rolling

Rémi Lafarge [1,*], Sebastian Hütter [2], Thorsten Halle [2] and Alexander Brosius [1]

1   Chair of Forming and Machining Processes, Technische Universität Dresden, 01062 Dresden, Germany; alexander.brosius@tu-dresden.de
2   Institute of Materials and Joining Technology, Otto-von-Guericke University, 39104 Magdeburg, Germany; sebastian.huetter@ovgu.de (S.H.); thorsten.halle@ovgu.de (T.H.)
*   Correspondence: remi.lafarge@tu-dresden.de; Tel.: +49-351-463-38701

**Abstract:** International objectives towards improved resource and energy efficiency require new manufacturing processes, such as the proposed thermomechanical tangential profiled ring rolling process. A rapid cooling-down for microstructural adjustment and a final calibration step via cold forming are combined into one single step. The reduction of process steps and the reduced number of heating cycles present opportunities for improved energy efficiency compared to traditional processes. Based on a series of experiments, this paper aims at showing the potential of this new process in terms of obtainable hardness and microstructure. The influence of the active cooling system and the rolling feed is demonstrated. They are identified as essential with regard to both the geometry and the microstructural properties of the produced part. Finally, the repeatability of the process is analyzed, and potential disturbances are identified and ranked.

**Keywords:** ring rolling; thermomechanical processes; energy efficiency

## 1. Introduction

The fast technological development of e-mobility vehicles (e.g., bikes, cars, scooters, etc.) and their specialized drive components poses major challenges for the corresponding production technologies. While in the past, the focus in forming technologies was the improvement of geometric tolerances, strength and stiffness, nowadays electrical, magnetic, and thermal properties have become increasingly relevant. Additionally, sustainability for energy-intensive forming technologies is an important issue that must be addressed. While the aforementioned properties can be adjusted via the microstructure of the material, sustainability can only be promoted by improving resources and process efficiency. Thermomechanical ring rolling could present a good example of a process where the desired geometrical and microstructural properties can be obtained while retaining a limited environmental footprint.

Ring rolling is an incremental forming process where a ring-shaped blank is formed using at least two tools, namely the main roll and mandrel. This process is used in a wide variety of applications, with ring sizes varying between a few centimeters and a few meters. Allwood et al. [1] present a comprehensive overview of hot and cold process variants and their applications. In a subsequent paper on incremental bulk forming, Groche et al. [2] provide a detailed analysis of ring rolling challenges, methods, and opportunities with regard to obtainable shape, simulation methods, and composite parts. Tangential profiled ring rolling (TPRR) is a subfamily of this process [3] aimed at the production of small- to medium-sized parts, typically a few centimeters to a few decimeters in diameter [4]. Its main application is the production of outer and inner bearing rings or similarly shaped parts. The process can be achieved either cold or hot, with the latter offering improved formability. In the production of ball bearings, TPRR has been proven to present a number of advantages, including a very short process time and high material efficiency compared to

cutting [5]. Profiroll GmbH, a producer of ring rolling machines, claims a reduced material use of up to 20% compared to cutting [6]. Furthermore, some studies [7] have reported an improvement in part performance and fatigue resistance due to a better fiber orientation in the produced part, similar to the improved performance of forged parts.

Following the typical manufacturing strategy, the TPRR process is carried out at room or semi-hot temperature with the aim of obtaining a near-net-shaped part, followed by a heat treatment such as quenching and tempering. In contrast, the concept of thermomechanical TPRR, recently introduced in Brosius et al. [8], is to combine an initial hot forming step with a rapid cooling-down for microstructural adjustment and a final calibration step via cold forming in one single process, as seen Figure 1. The thermomechanical TPRR of small rings presents a good opportunity for such a combined thermomechanical process, as the ratio of surface to volume of the part is relatively high and the forming forces even in the cold state are relatively small. This enables a very rapid and homogeneous cooling of the parts, which allows for targeted changes in microstructure and, subsequently, hardness [9].

The bulk of the energy consumption and direct $CO_2$ emission in hot forming processes is heating (scope 1 emissions) or upstream energy production if an electricity furnace is used (scope 2); therefore, reducing heating cycles is a very efficient way to improve sustainability [10]. Furthermore, the enhanced material efficiency helps to reduce the upstream emissions (scope 3).

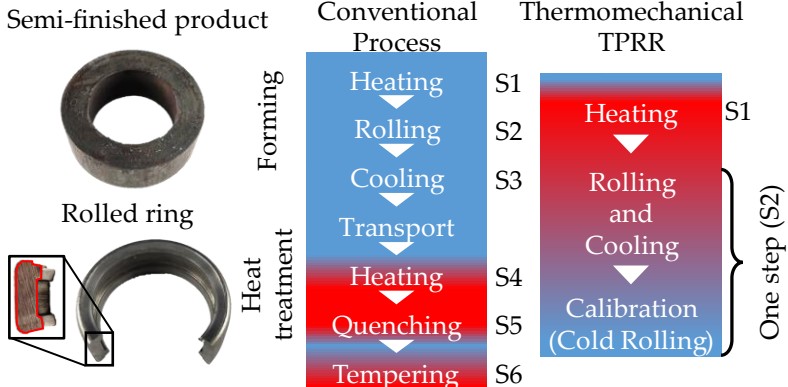

**Figure 1.** Thermomechanical TPRR as introduced in [8].

## 2. Materials and Methods

TPRR experiments were realized on a retrofitted UPWS 31,5.2 from VEB Werkzeugmaschinenfabrik Bad Düben (1986), now known as Profiroll Technologies GmbH. Figure 2 presents the different parts of the machine.

During the process, the main roll (C) is driven by an electric motor, while a hydraulic cylinder pushes it towards the free-rolling mandrel. The combined action of the translational and rotational movement of the main roll leads to the forming of the ring (F). The guide rolls (B) are pushed toward the ring by two hydraulic cylinders and help stabilize the forming process.

The contact areas between the tools and the blank are a distinct heat sink and lead to significant cooling during the process. For greater control over the cooling rate and additional cooling potential, the machine has been equipped with a compressed air-cooling system (A). A proportional valve controls the airflow.

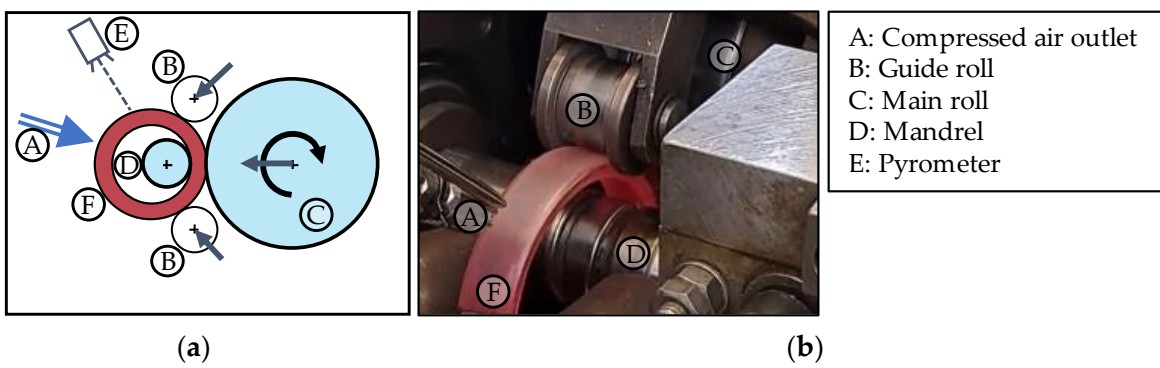

|  |
|---|
| A: Compressed air outlet |
| B: Guide roll |
| C: Main roll |
| D: Mandrel |
| E: Pyrometer |

(**a**)　　　　　　　　　　　　　　　　　(**b**)

**Figure 2.** Diagram (**a**) and photo (**b**) of the TPRR machine.

### 2.1. Experimental Setup

The tool geometry corresponds to the geometry of the outer ring of a roller bearing, as presented in Figure 3. The main roll and the mandrel have an external diameter of 245 mm and 45 mm, respectively.

In this study, the rings are made of a 16MnCr5 steel (1.7131, AISI 5115). Its hardenability is limited by its relatively small carbon content, but hardness higher than 400 HV can be obtained. Its high formability and low yield stress make this alloy a good compromise for the demands of combined warm–cold forming in thermomechanical TPRR. The blanks are cut from a seamless, normalized steel tube.

The rolling machine is equipped with different sensors to determine the main roll displacement, ring growth, surface temperature, rolling force, and airflow in the active cooling system. The sensors are connected to an HBM QuantumX data acquisition device.

For measuring the temperature, an Optris CT3MH1 pyrometer is used. The emissivity value was obtained by comparison with measurements of the temperature using a Type K thermocouple at different temperatures. A dispersion of 8–10% of the emissivity has been observed, mainly depending on oxidation and temperature. Furthermore, the temperature of the tool was measured at the beginning of each experiment using a handheld pyrometer.

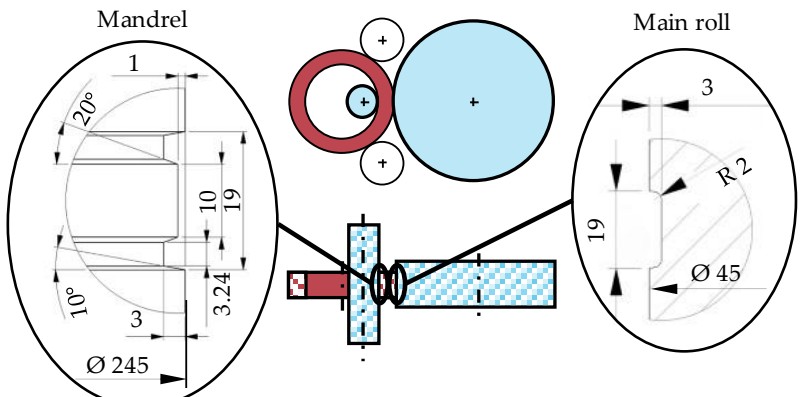

**Figure 3.** Geometry of the profile of the tools.

### 2.2. Process Window of Stability

The choice of an adequate stabilization force is necessary for process stability. In cold TPRR, if the value is too low, the rolling process becomes unstable, generating highly asymmetric ring deformation and production of scrap parts (see Figure 4A). However, if the force is too high, the ring might be permanently deformed (see Figure 4B) or, in the most severe cases, break (see Figure 4C). This issue is also present in thermomechanical ring rolling, with the added challenge that the temperature influences the current mechanical properties of the ring, creating a continuously changing process window. Therefore, the

maximum and minimum allowed stabilization forces are dependent on the current ring temperature. A schematic representation of the process window development depending on the current temperature is shown in Figure 4, which indicates that the size of the usable process window is reduced for higher ring temperatures. The stabilization force is given in arbitrary units representing the hydraulic pressures influencing the stabilizing roll as it is not possible to measure it with the current machine setup. In the following part, the stabilization force was selected so that a stable process was completed in the full temperature range of 900–25 °C.

A low main roll feed rate below 0.1 mm/s has also been found to be detrimental to process stability. A high feed is correlated with an increase in the forming force, which carries a risk of exceeding machine capability. Furthermore, high feed rates are known to cause non-circularity faults in ring rolling [11].

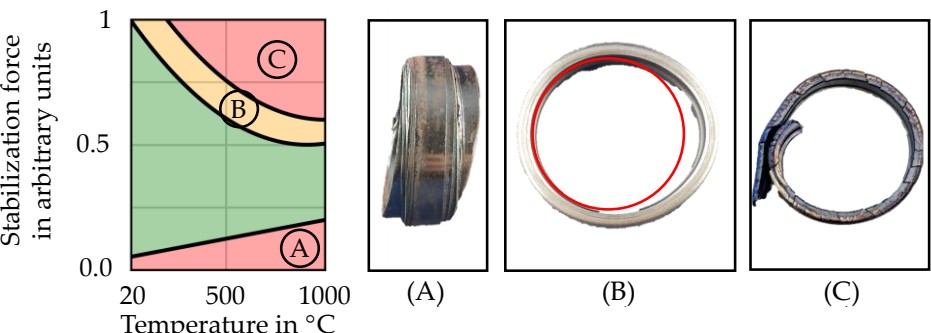

**Figure 4.** Acceptable value of the stabilization force with example of scrap rings, with example of scrap rings (**A**–**C**).

### 2.3. Process Parameters and Description

The rings are heated in a furnace set to a temperature of 900 °C for 1800 s to ensure complete austenization. The standard deviation of the austenization time is estimated to be smaller than 30 s.

The operator positions the warm ring at the mandrel and closes the tools manually before the process is started. Because of the manual transfer, there is a significant variation of the transfer time. The average transfer time is 25 s with a standard deviation of 8.25 s (measured over 20 experiments). Due to their low mass and high surface-to-volume ratio, the ring experiences relevant cooling during transfers (around 5 K/s).

The main roll feed is selected by opening and closing the valve controlling the hydraulics driving the main roll. The pressure in the cylinder is limited by a release system, which corresponds to a force of around 50 kN. When the process is started, the main roll advances towards the mandrel until the force reaches this maximum. Acceptable values for the main roll feed rate have been found in the range of 0.5–1.2 mm/s. The main roll rotational speed is fixed at 100 rotations per minute.

The airflow in the active cooling system can be set between 0 and 500 L/min, using a linear proportional valve. A value of up to 600 L/min can temporarily be achieved.

### 2.4. Design of Experiments

An appropriate design of experiments (DOE) must be chosen in order to answer the two main questions for this investigation: Firstly, what is the influence of the two main parameters—the main roll feed and the air-cooling flow? Secondly, what disturbances influence process repeatability? The first answer can be given using a screening experiments design, while the second requires repetitions with the same parameters. Therefore, two sets of experiments have been carried out: firstly, a full factorial, four-level plan; and secondly, 12 repetitions.

In total, 36 ring rolling experiments have been realized. Table 1 presents the chosen parameters. To ensure unbiased results, experimental plans were randomized, consequently

limiting the influence of tool temperature change or wear. All experiments were completed in two batches (one for the screening, one for the repeatability evaluation), with a fixed cadence of one ring every 10 min to ensure minimal variation of the initial conditions. Finally, before both series, three "warm-up" rings were rolled but not included in the analysis. The temperature of the tools rises during the process, and they cool down during the dead time. Because of residual heat at the end of the dead time, the start temperature of the tools slowly increases for the first few experiments. The warm-up series is designed to reach a steady state with regard to the start tool temperature.

**Table 1.** Summary of the parameters used for the process.

| Series Name | Experiments IDs | Airflow Value in L/min | Mandrel Feed Control Value | Corresponding Theoretical Feed |
|---|---|---|---|---|
| Preliminary work (ASK) [9] | ASK_A–ASK_G | 0 up to 600 | 4.5 up to 8.5 | 0.15 up to 1.0 |
| Repeatability (Rep.) | A1–A12 | 300 | 6 | 0.3 |
| Exploration (Exp.) | B–R | 0 up to 500 | 5.5 up to 8 | 0.25 up to 0.75 |

*2.5. Hardness Measurement*

Each ring produced according to the scheme described above was left to rest until cooled to room temperature. Next, both the inner and outer diameter were measured, using multiple measurement positions to account for possible ellipticity.

Next, a section from each ring was cut. This section was then prepared for metallographic investigation (see Figure 5). The samples were mounted in resin, then ground and polished to a surface roughness of 3 μm. Vickers HV1 micro-hardness measurements were performed on a Buehler Wilson VH3300. In each ring, a line profile of 10 points near the center line of the ring was taken. It was found that the hardness does not vary with position; therefore, the hardness reported refers to the average. Additionally, micrographs were taken at 50× and 200× magnification using a Zeiss Axiovert 200M incident light microscope and were used to determine grain size (see Figure 5).

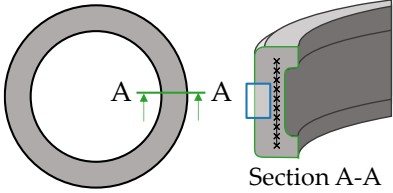

Section A-A

**Figure 5.** Sample preparation for optical micrography (blue box) and hardness measurement (black points and line).

## 3. Results

*3.1. Process Phases*

Figure 6 presents the results of a ring rolling experiment (experiment A1, see Table 1), including the different process phases. The forming phase starts when the space between the main roll and the mandrel is smaller than the initial thickness of the ring. It ends when the main roll surface is in contact with that of the mandrel. During the active cooling phase, the airflow remains enabled, but the feed rate is zero. The rolling force remains high because the main roll is pushed against the mandrel, and the rolling moment is much smaller during this phase, as no forming occurs. The bumps in the temperature at the end and the beginning of the process are caused by uncontrolled movement of the ring as the tool opens or closes.

From the sensor's output, a few values can be computed. Firstly, the average tangential plastic strain can be estimated using the following equation:

$$\varepsilon_{\theta\theta}^{p} = \ln\left(\frac{d - d_0}{d_0}\right) \tag{1}$$

where $d$ is the current diameter in mm, and $d_0$ the starting diameter (65 mm). This equation neglects the elastic and thermal strains, but they are a few orders of magnitude smaller. Secondly, assuming an exponential decay of the temperature, a time constant is determined using the following:

$$\theta_R = \left(\theta_R\left(t = t_{0,phase}\right) - \theta_{\text{Room}}\right) \times \exp\left(-\frac{t}{\tau_{phase}}\right) + \theta_{\text{Room}} \tag{2}$$

where $\theta_R$ is the temperature of the ring, $t_{0,phase}$ the time at which the phase begins, and $\tau_{phase}$ the time constant. This computation is made using a log regression for both the forming and the active cooling phase, and the time constants are then noted as $\tau_f$ and $\tau_{ac}$, respectively.

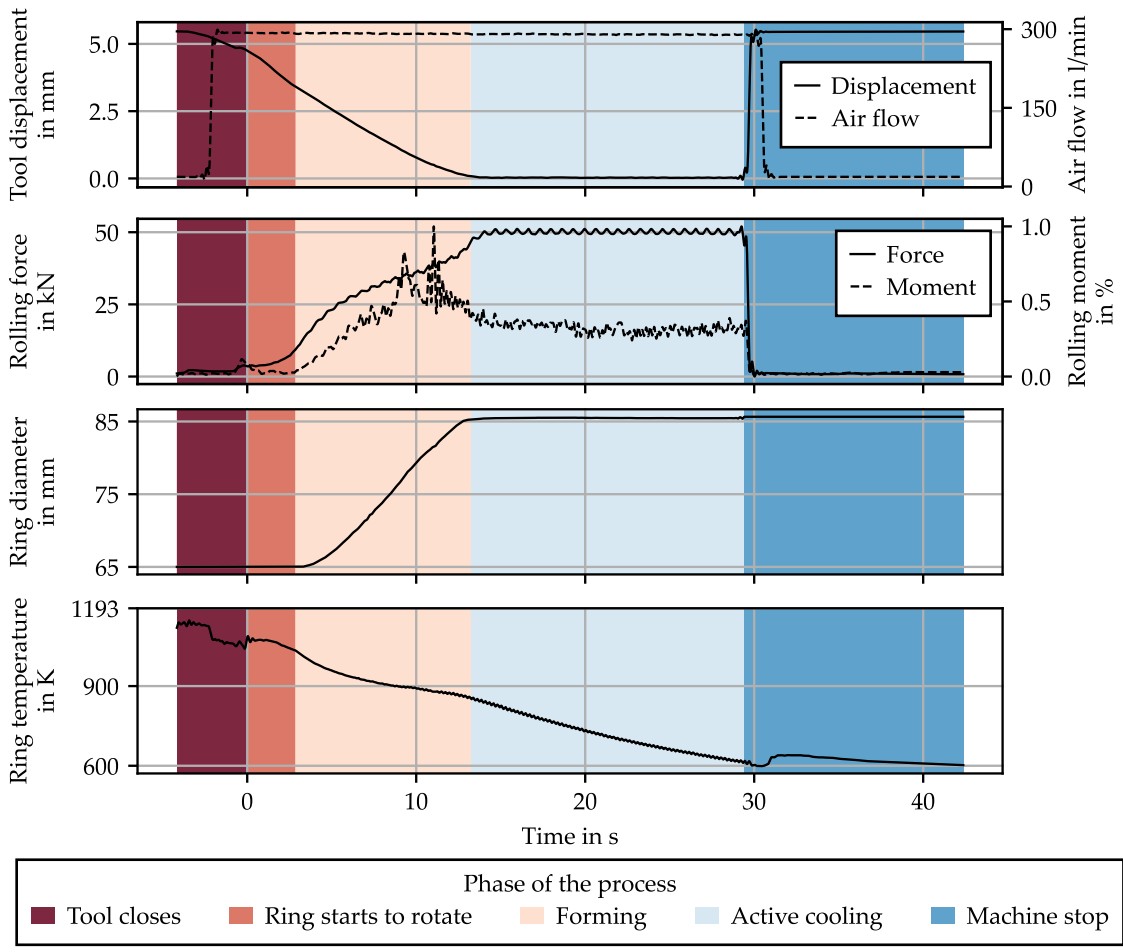

**Figure 6.** Time series of different sensor signals during the different process phases for experiment A1.

### 3.2. Process Window Screening

Figure 7 summarizes the obtained results. It is apparent that a wide range of final diameters and hardness can be achieved by changing both the feed rate and airflow. A moderate correlation (Pearson correlation: $\rho = 0.73$, associated *p*-value: $p = 7.9 \times 10^{-7}$) is observed between the final hardness and the final diameter.

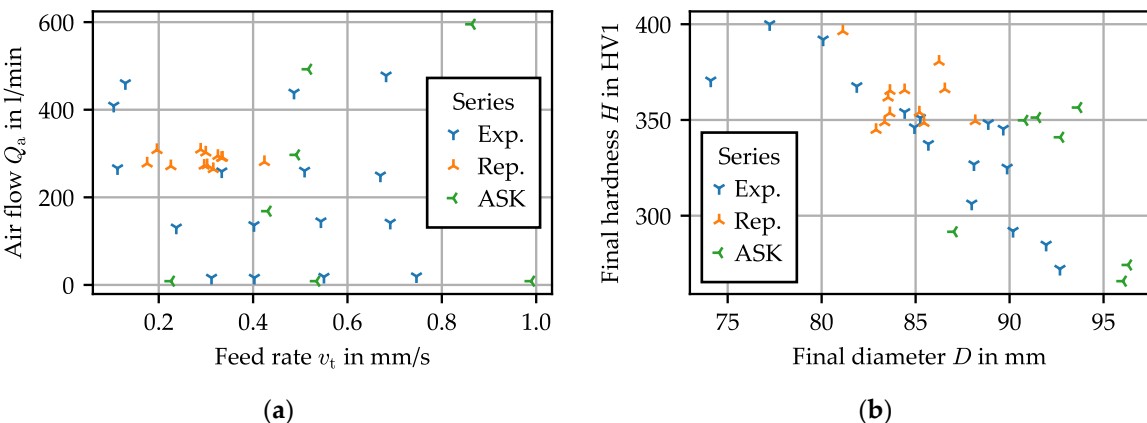

**Figure 7.** Overview of the experimental parameters (**a**) and results (**b**).

### 3.2.1. Influence of Process Parameters on the Ring Geometry

The first goal of the process is to achieve a given geometry. In cold TPRR, the ring size and tool geometry is computed to achieve a given size. Because of the strong temperature change in thermomechanical TPRR, the final diameter is not only dependent on the blank size and tool geometry. During the experiments, a noticeable correlation between the feed rate and diameter was obtained ($\rho = 0.74$, $p = 9.4 \times 10^{-4}$), while the airflow influence is much weaker but still significant ($\rho = -0.36$, $p = 0.032$).

An attempt to model the diameter as a function of feed rate and airflow is made, by means of a regression on the data at disposition, using the following model:

$$D = \frac{A_\mathrm{D}}{1 + \exp(D_\mathrm{tr})} + B_\mathrm{D} \tag{3}$$

$$D_\mathrm{tr} = k_{\mathrm{D},1} \times v_\mathrm{t} + k_{\mathrm{D},2} \times v_\mathrm{t}^2 + k_{\mathrm{D},3} \times Q_\mathrm{a} + k_{\mathrm{D},4} \times Q_\mathrm{a}^2 + k_{\mathrm{D},5} \times v_\mathrm{t} \times Q_\mathrm{a} + k_{\mathrm{D},6} \\ \times \log(v_\mathrm{t}) + k_{\mathrm{D},7} \times Q_\mathrm{a}^{\frac{1}{2}} + k_{\mathrm{D},8} \times \log(v_\mathrm{t}) \times Q_\mathrm{a}^{\frac{1}{2}} \tag{4}$$

where $D$ is the final ring diameter, $D_\mathrm{tr}$ the diameter transformed by the sigmoid function, $Q_\mathrm{a}$ the airflow, $v_\mathrm{t}$ the feed rate, and $k_{\mathrm{D},i}$ the parameters of the model. The ring diameter is, however, contained within a given range; for example, the final diameter cannot be smaller than the start diameter. To enforce maximal and minimal values of the predicted diameter, the Sigmoid function is used, as defined in Equation (3). $A_\mathrm{D}$ and $B_\mathrm{D}$ therefore represent the physical limits of the model and are fixed at 65 mm and 55 mm, respectively, limiting the modelled value of the diameter between 65 and 120 mm. The model is fitted using the ordinary least square algorithm of the Python package statsmodels. All the statistically insignificant factors (with a $p$-value higher than 5%) are then dropped.

Equation (4) of the model then simplifies to the following:

$$D_\mathrm{tr} = k_{\mathrm{D},6} \times \log(v_\mathrm{t}) + k_{\mathrm{D},8} \times \log(v_\mathrm{t}) \times Q_\mathrm{a}^{\frac{1}{2}} \tag{5}$$

The obtained coefficient of determination, $R^2$, is 0.775, and the probability associated with the F-statistic for the model is $4.3 \times 10^{-11}$. The $p$-value of each factor is below 0.010. Figure 8 represents the model and its comparison with the experiments.

An increase in feed rate results in an increase in the ring diameter. However, airflow has the opposite effect: if the ring is cooled with a higher airflow and resulting higher cooling rate, the final diameter is smaller. This effect is quite noticeable, and no explanation could be found in the literature relating to TPRR. There is a strong link between the forming temperature and the growth rate of the ring, as illustrated in Figure 9. Two facts support this claim. Firstly, the average temperature during forming is positively correlated with

ring diameter ($\rho = 0.88$, $p = 1.4 \times 10^{-12}$). Secondly, as the temperature falls, the ring growth rate decreases. Two hypotheses can explain this effect:

- Friction can be affected by temperature, and in return influence ring growth. The tangential friction stress is limited to the yield shear stress of the material. This value is often attained in hot and semi-hot forming [12]. The yield stress being strongly dependent on temperature, this could change the flow of the metal in the forming zone, resulting in a change of diameter.
- For the ring radius to grow, the ring curvature at each point must change: as the ring elongates in the deformation zone, the rest of the ring must bend. This effect has been modeled as an elastic bending of the ring in semi-analytical models [13], implying that the reaction force to curvature change is directly proportional to the elastic modulus. As the ring cools down, the Young's modulus of the material increases, and so does the tangential stiffness. The higher stiffness limits the ability of the material to flow in the tangential direction within the forming zone.

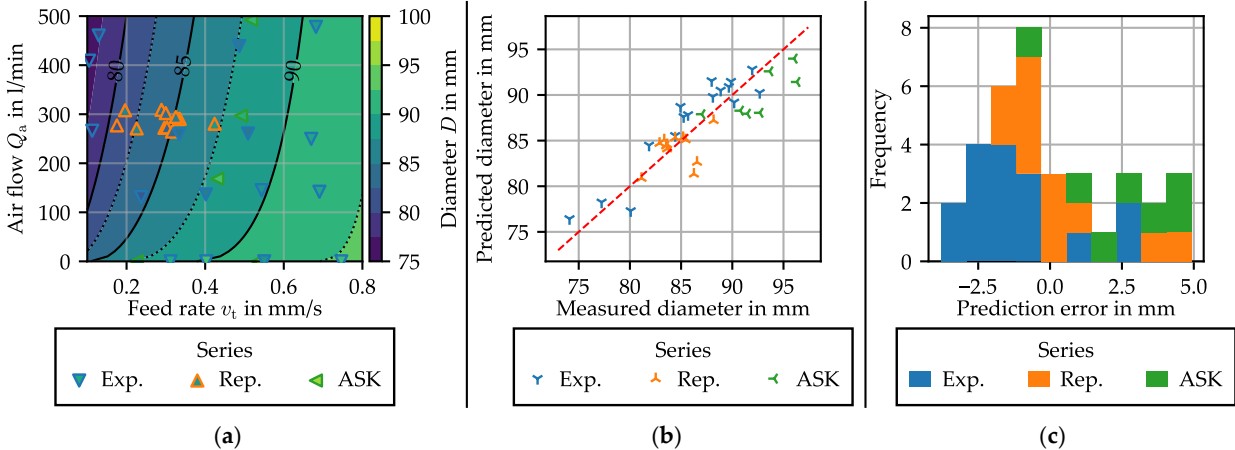

**Figure 8.** Model of the diameter as a function of process parameters: (**a**) predicted diameter as a function of process parameters, the colors inside each point representing a measured value; (**b**) correlation between measured and predicted diameter; (**c**) histogram of the model error.

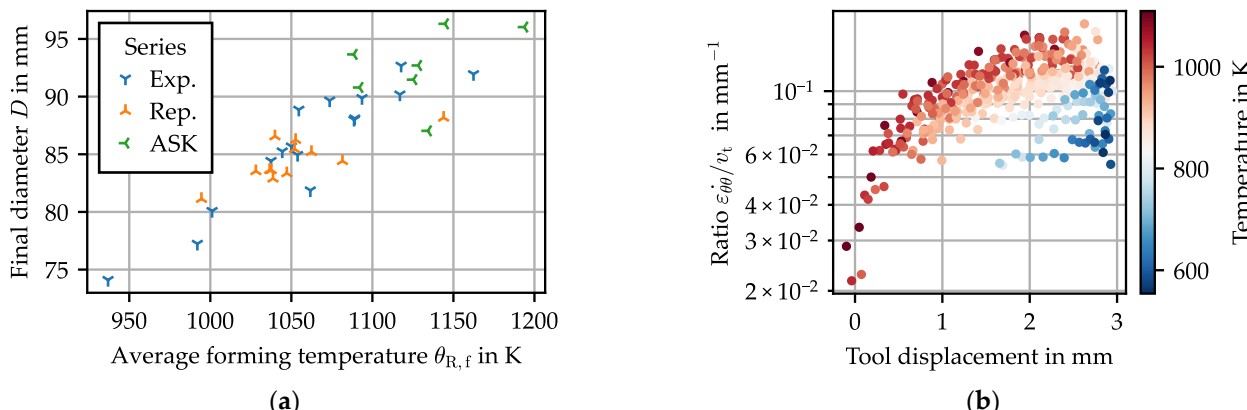

**Figure 9.** Influence of temperature on ring growth: (**a**) correlation between average forming temperature and final diameter; (**b**) link between temperature and tangential plastic strain rate, based only on data sets "Exploration" and "Repeatability".

### 3.2.2. Influence of the Process Parameters on the Ring Hardness

The second objective of the process is to obtain a given hardness for the produced ring. Steel hardness is a result of many parameters including phase proportion, grain size,

and dislocation density. Because of the strong temperature change in the process and the high cooling rate, it is expected that phase proportion and phase change would be the dominating influences.

A first overview can be obtained from the micrographs shown in Figure 10, which support this assessment. Depending on process conditions, varying compositions of ferrite, perlite, and even bainite can be achieved. At the same time, the resulting grain size is influenced by both the plastic deformation history and the temperature, especially in the temperature range critical for recrystallization.

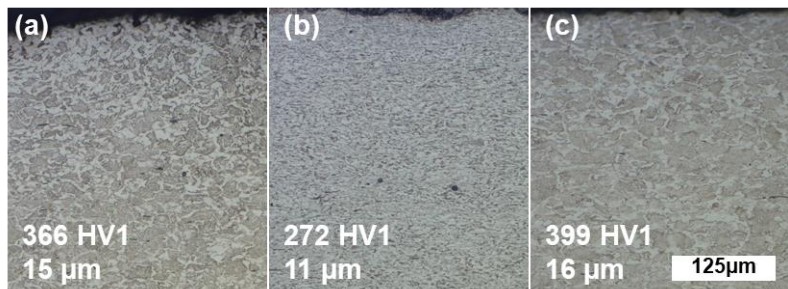

**Figure 10.** Representative cross-section micrographs of some achievable microstructures and associated hardness and average grain size: mid-range hardness with larger grain size and ferritic-perlitic microstructure (**a**), much lower hardness in spite of lower grain size due to mostly ferritic microstructure (**b**), and high achievable hardness due to partially bainitic microstructure (**c**).

This is confirmed by the strong correlation ($\rho = -0.86$, $p = 1.6 \times 10^{-5}$) obtained between the cooling time from 800 °C and 400 °C $t_{8-4}$ and hardness. Further investigations are necessary to verify the influence of the forming on the resulting hardness, but the cooling rate seems to be the most influential factor there.

By applying the same procedure as for the diameter in the previous section, the following simplified model is obtained:

$$H = \frac{A_{\mathrm{V}}}{1 + \exp(H_{tr})} + B_{\mathrm{V}} \tag{6}$$

$$H_{\mathrm{tr}} = k_{\mathrm{H},1} \times v_{\mathrm{t}} + k_{\mathrm{H},3} \times Q_{\mathrm{a}} + k_{\mathrm{H},6} \times \log(v_{\mathrm{t}}) + k_{\mathrm{H},7} \times Q_{\mathrm{a}}^{\frac{1}{2}} \tag{7}$$

where $H$ is the hardness, $H_{\mathrm{tr}}$ the hardness transformed by the sigmoid function, $Q_{\mathrm{a}}$ the airflow, and $v_{\mathrm{t}}$ the feed rate. $A_{\mathrm{V}}$ and $B_{\mathrm{V}}$ are fixed to 270 HV1 and 150 HV1, respectively, limiting the modelled hardness value between 150 and 420 HV1. The obtained $R^2$ is 0.87, and the probability associated to F-statistic for the model is $7.16 \times 10^{-3}$. The $p$-value of each factor is below 0.026. Figure 11 represents the model results.

The influence of the airflow $Q_{\mathrm{a}}$ is to be expected: the airflow very significantly accelerates the cooling rate, as is shown in Figure 11. As the airflow increases, so does the air speed at the surface of the ring, which increases the forced convection. As the cooling rate increases, so does the hardness, due to the change in proportion of hard phases. Furthermore, Figure 12 clearly shows that cooling appears to be slightly slower during the forming phase than during the active cooling phase. That is probably due to additional thermal energy added to the process generated by the plastic deformation. Another contributing factor is the increase in surface-to-volume ratio during the forming phase, which improves the efficiency of the forced convection cooling.

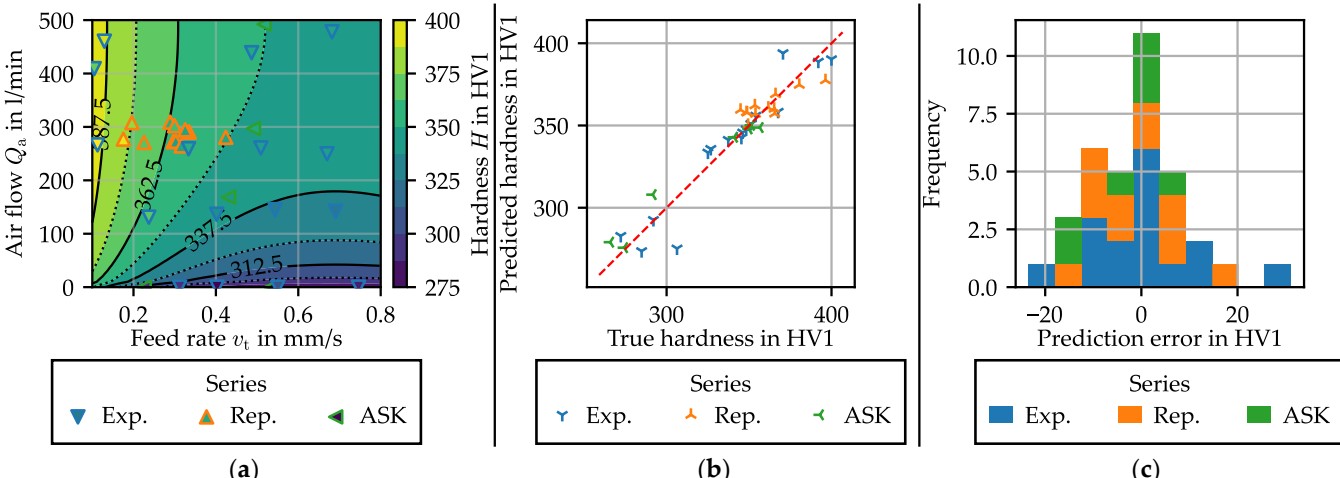

**Figure 11.** Model of the hardness as a function of process parameters: (**a**) predicted diameter as a function of process parameters, the colors inside each point representing a measured value; (**b**) correlation between measured and predicted diameter; (**c**) histogram of the model error.

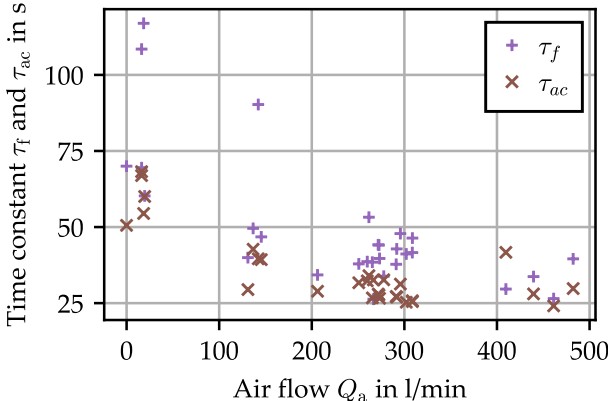

**Figure 12.** Cooling time constant during forming ($\tau_f$) and during active cooling ($\tau_{ac}$) as a function of airflow, computed as presented in Section 3.1. Data are extracted from data sets "Exploration" and "Repeatability".

The influence of feed rate on hardness cannot be definitively explained at this time, but the following hypotheses can be made:

- A slower feed rate means that the forming phase is longer. As the cooling rate is lower during this phase, this reduces the quantity of hard phase being formed during the process.
- A higher feed rate means a higher strain rate. Since forming begins at high temperature, this will affect the amount of recrystallization occurring during the early stages of forming. In this case, this will be mostly dynamic recrystallization triggered by the continuous production of dislocations during the forming process. This hypothesis is supported by a decrease of grain size with feed rate, as shown in Figure 13 ($\rho = -0.44$, $p = 0.02$). Following the Hall–Petch relation, this increases the measured hardness [14].

Further investigations and analysis of the rings, such as electron backscatter diffraction (EBSD) or X-ray crystallography, would help to find a definitive answer to that question.

The steel (16MnCr5) used in this investigation has only a limited hardenability; therefore, the value of hardness obtained here is of limited use from a technological standpoint. However, if the same cooling rate is applied to steel with a better quenchability, a higher value of hardness may be expected [9].

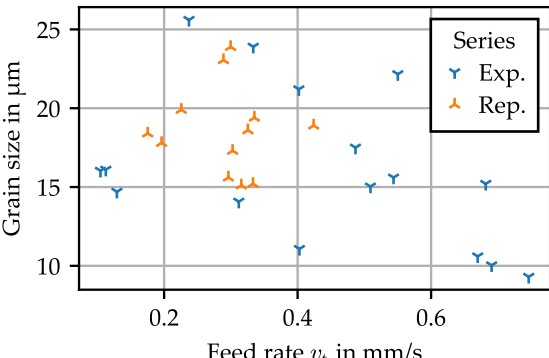

**Figure 13.** Constant average grain size as function of feed rate and air flow. Data are extracted from data sets "Exploration" and "Repeatability".

*3.3. Process Repeatability*

The combination of the forming and thermal treatment phase poses a challenge for process stability and repeatability. Indeed, by separating the processes into two phases, first forming, then hardening, the number of possible disturbances is reduced, and it is easier to limit their influence. In a combined process, the number of disturbances and their cross-effect may be more important.

Process capability is a function of the capability of the actuator of the machine and the sensitivity of the process to the various perturbations. The dataset "Repeatability" is used to have a first look at the possible disturbances and their effect on the process. The valves controlling the feed rate and the airflow are set at the same value, and 13 repetitions have been performed.

A significant variation is observed for both the hardness (average value: 360 HV1, standard deviation: 14 HV1, minimum: 345 HV1, maximum: 396 HV1) and the diameter (average value: 84.5 mm, standard deviation: 1.82 mm, minimum: 31.1 mm, maximum: 88.2 mm). The machine repeatability is quite limited; this is mainly due to a lack of control of the feed rate and airflow in the present setup. However, other external factors could influence the process output. Therefore, additional measurements have been made to identify potential disturbances:

- Tool surface temperature: During the process, both tools are heated through contact with the hot work piece. They cool down between the experiments. After a few experiments, an equilibrium is reached, which is dependent on the downtime between two experiments. Even after a few "warm-up" rings, there is always a variation of the tool temperature, albeit of lower magnitude. Because the thermal conduction between the tool and the ring is proportional to their temperature difference, this can influence the cooling rate of the ring and could therefore impact both the diameter and the hardness.

- Start temperature: Even assuming that the temperature in the furnace is homogeneous and constant, there is still a possibility of having a significant variation of the start temperature. Indeed, the transfer of the part, the closure of the mandrel, and the machine startup is realized manually. During this time, the part is cooling down at around 5 K/s.

Other perturbation sources are possible, such as tool wear or variation in material properties, but their effect is probably much smaller than those previously discussed.

Figure 14 shows the influence of both process parameters and potential disturbances. To test and quantify the influence of each variable on the outcome value, the standardized regression coefficients are computed. This is realized by fitting a linear model matching each regressed value (in this case, $D$ and $H$) to the wanted parameters, after having standardized all the data, which means setting the standard deviation to one.

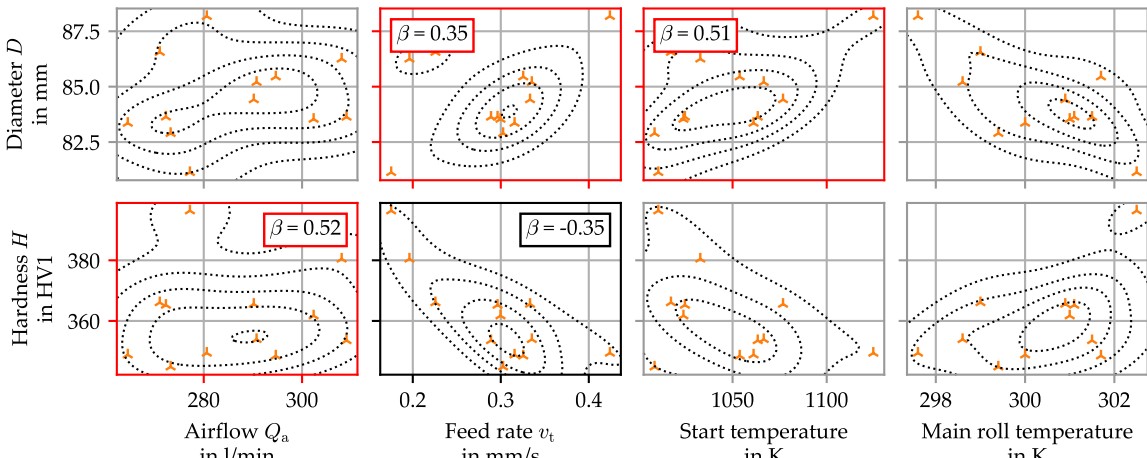

**Figure 14.** Influence of different parameters on the process repeatability. The data are superposed with kernel density estimates, represented as dotted lines. The standardized regression coefficients (*β*) associated with a *p*-value lower than 5% are shown in a red frame, while those associated with a *p*-value inferior to 10% are shown in black.

The linear regression gave an adequate model, with R-value of 0.748 and 0.84 for the diameter and the hardness regression, respectively. The standardized values associated with a *p*-value inferior to 5% are shown in a red frame, while the ones associated with a *p*-value inferior to 10% are shown in a black frame.

The hypothesis on the influence of the tool temperature can be rejected. The start temperature seems to impact only the diameter, but not the hardness. This may be because the start temperature remains high compared to the microstructural transformation temperature. The feed rate has a very high standard deviation, mainly because of the poor repeatability of the ring rolling machine, which is equipped with an uncontrolled hydraulic system. However, it is never the major influencing factor on both target values. Finally, the influence of the airflow variation on the hardness is significant. The deviation of the airflow is caused by variation in the air pressure provided to the ring rolling machine.

## 4. Conclusions and Perspectives

Thermomechanical TPRR is still a new process, and according to the literature, no ring rolling process has ever been combined with active cooling to achieve specific hardness. Despite early work on the possibility of combining forming and hardening operation, a study of the process capability of this concept is still needed.

In this study, the influence of the major process parameters was investigated. It has been shown that both process parameters (feed speed and airflow) influence both target values (diameter and hardness). This means that the process must be seen as a multiple-input, multiple-output (MIMO) system. Possible explanations were provided to demonstrate the cross-influence. The effect of the temperature on the diameter was explained and is supported by measurements. The influence of the feed rate on the cooling rate was also presented.

Finally, the repeatability of the process was studied. Potential disturbances have been identified and ranked. It is concluded that the capability of the machine must be enhanced by using closed-loop control on the airflow and the feed rate. However, the influence of the forming temperature needs to be compensated by other means. This can be done using model predictive control (MPC), as introduced in a previous publication on the subject of thermomechanical ring rolling [15].

Overall, the concept of this process has been validated. Thermomechanical TPRR has the potential to be an innovative method to produce a hardened ring-shaped part in a more energy-efficient way. However, some challenges for future investigations have also been highlighted:

- Obtaining a sufficient process capability will require advanced control systems, for each machine actuator but also for the process itself, using model process control techniques.
- As in every hot or semi-hot forming operation, tool wear might become a challenge, especially when harder material is used.
- The results presented in this paper need to be confirmed using material with a better hardenability, such as 100Cr6 steels. The suitability of the process for harder steel must be validated, especially when some forming occurs at lower temperature.

**Author Contributions:** Conceptualization, R.L.; formal analysis, R.L. and S.H.; funding acquisition, T.H. and A.B.; investigation, R.L.; project administration, T.H. and A.B.; supervision, T.H. and A.B.; writing—original draft, R.L. and S.H.; writing—review and editing, A.B. All authors have read and agreed to the published version of the manuscript.

**Funding:** The authors would like to thank the German Research Foundation DFG for their support of this research within the priority program 'SPP2183' under grant numbers 'BR 3500/23-2' and 'HA5209/11-2'.

**Data Availability Statement:** The data supporting this study's findings, that are not already presented in the article, are available from the corresponding author upon reasonable request.

**Conflicts of Interest:** The authors declare no conflict of interest.

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
