# Peer review of "Process Window and Repeatability of Thermomechanical Tangential Ring Rolling"

_jmmp, doi:10.3390/jmmp7030098_

Round 1
Reviewer 1 Report
This paper investigated the hardness and microstructure information of the samples prepared by a thermomechanical TPRR process, focusing on the influence of major process parameters. Some useful information has been provided for industrial applications. The following suggestions are provided for further improvement:
1) Some typos and languages existed in this paper. For example, "TRPP" should be TPRR. Please read through the whole paper to correct these issues. By the way, what is the "DOE"?
2) For the microstructure results, no microstructure images were provided. The authors claimed that optical micrography was conducted. So please provide some images for a clear illustration.
3) It is strange that no "Conclusion" part was set for this paper. I suggest the "Discussion" part to be shortened, and described as "Conclusion and Prospectives".
Some language issues exist, please read through the whole paper to correct these issues.
Author Response
Dear reviewer,
Thank you for the thorough review of our paper. We have modified it according to your remarks and advices. Below, you will find our points by points answer.
Remark 1: Some typos and languages existed in this paper. For example, "TRPP" should be TPRR. Please read through the whole paper to correct these issues. By the way, what is the "DOE"?
A complete proof reading of the paper was done and mistakes were corrected. DOE means Design of Experiments; the acronym is now explicitly detailed in the paper.
Remark 2: For the microstructure results, no microstructure images were provided. The authors claimed that optical micrography was conducted. So please provide some images for a clear illustration.
Optical micrography of tree representing samples have been included in the paper, Figure 10, page 9 of the paper.
Remark 3: It is strange that no "Conclusion" part was set for this paper. I suggest the "Discussion" part to be shortened, and described as "Conclusion and Prospectives".
This was done in accordance to the recommendation of the journal templates. We modified the part slightly and renamed it according to your recommendation.
Best regards,
Rémi Lafarge
Reviewer 2 Report
1. Figure 1 illustrates the conventional process and the thermomechanical TPRR. As shown in Fig. 1, the conventional process is composed of forming and heat-treatment, and thus it fits for almost all steels. In contrast, the final calibration in the TPRR is performed by cold forming (Note: “cold colling” in One step (S2) should be “cold forming”?), and thus this process maybe not suitable to some hard steels. The effective range of the proposed TPRR should be described.
2. 16MnCr5 steel is a widely used steel. Probably, this steel (or similar steels) is used to produce rings by Conventional Process, i.e., maybe there is experimental data in literature. The advantages of the thermomechanical TPRR are listed in Lines 49-54. For the 16MnCr5 steel, if is it possible that the merits and demerits of the thermomechanical TPRR are quantitatively analyzed by comparing the existing data with the present study?
Author Response
Dear reviewer,
Thank you for the thorough review of our paper. We have modified it according to your remarks and advice. Below, you will find our points by points answer.
Remark 1: Figure 1 illustrates the conventional process and the thermomechanical TPRR. As shown in Fig. 1, the conventional process is composed of forming and heat-treatment, and thus it fits for almost all steels. In contrast, the final calibration in the TPRR is performed by cold forming (Note: “cold colling” in One step (S2) should be “cold forming”?), and thus this process maybe not suitable to some hard steels. The effective range of the proposed TPRR should be described.
The last forming step of calibration, corresponding to the forming that occurs at the end of the process is realized at a relatively low temperature ‑ below the recrystallization temperature of around 700K in our case ‑ which categorizes it as a cold forming step. However, the final forming is completed well above room temperature. The suitability of the process for harder steels, such as a 100Cr6 for example, is a question that still needs to be addressed in future investigations. This limitation is now highlighted in the conclusion, in lines 398-401.
Remark 2: 16MnCr5 steel is a widely used steel. Probably, this steel (or similar steels) is used to produce rings by Conventional Process, i.e., maybe there is experimental data in literature. The advantages of the thermomechanical TPRR are listed in Lines 49-54. For the 16MnCr5 steel, if is it possible that the merits and demerits of the thermomechanical TPRR are quantitatively analyzed by comparing the existing data with the present study?
To our knowledge, there is no such collection of data available in the literature. The complete process route (cold ring rolling followed by heat treatment) is rarely investigated. A notable study was realized by Lu et al. [1] but for a completely different kind of steel, equivalent to a 100Cr6. The same team has realized another study of heat treatment following cold ring rolling, but again for a steel with a much higher carbon content.
Therefore, such a quantitative comparison is not possible at this time. We, however, think that such a study could be of interest for future investigations.
Best regards,
Rémi Lafarge
[1] B. H. Lu, L. Hua, X. H. Han & G. H. Zhou (2016) Microstructure evolution of GCr15 in cold ring rolling and following heat treatment, Materials Science and Technology, 32:16, 1702-1711, DOI: 10.1080/02670836.2016.1142703.
Round 2
Reviewer 2 Report
The authors well revised the manuscript.